# Current Concepts and Challenges in the Treatment of Cleft Lip and Palate Patients—A Comprehensive Review

**DOI:** 10.3390/jpm12122089

**Published:** 2022-12-19

**Authors:** Anna Paradowska-Stolarz, Marcin Mikulewicz, Irena Duś-Ilnicka

**Affiliations:** 1Division of Dentofacial Anomalies, Department of Orthodontics and Dentofacial Orhopedics, Wroclaw Medical University, Krakowska 26, 52-425 Wrocław, Poland; 2Oral Pathology Department, Wroclaw Medical University, ul. Krakowska 26, 52-425 Wrocław, Poland

**Keywords:** cleft lip, cleft palate, dental care, orthodontics

## Abstract

Cleft lip and cleft palate has one of the highest incidences in the malformations of the oral cavity, that varies between populations. The background underlying the issue of cleft lip and palate is multifactorial and greatly depends on the genetic factors and environmental factors. The aim of this nonsystematic narrative review is to present the cleft palate and or lip pediatric population as target for interdisciplinary treatment. The purpose of this narrative review is to sum up the modern knowledge on the treatment of patients with clefts, as well as to highlight the importance of the great need for cooperation between different dental specialists along with medical professionals such as oral surgeons, prosthodontists, orthodontists along with medical professions such as pediatricians, speech therapists and phoniatrics, and laryngologist.

## 1. Introduction

Cleft lip and/or palate (CLP) is one of the most common malformations, with a majority of isolated types, observed in ca. 1.7 per 1000 live births. The etiology of cleft remains unclear, although the combination of genetic and environmental factors is indisputable. In addition to the genetic factors, epidemiology considers environmental factors, the highlighting of drugs, alcohol and smoking. As well as the folic acid intake of the mother. It has been observed that the frequency of cases varies according to each race in a given country. In most cases, the deformity is unilateral, but a bilateral version is also possible [1,2,3].

CLP is a congenital deformity, and the malformation occurs in the womb. Normally, palatal shelves draw nearer and fuse in the 5th to 12th week while in utero. The disturbance in mesenchymal and endodermal cell proliferation in embryogenesis guides to disturbances in development that may manifest as clefts [4]. The division of clefts includes cleft palate and cleft lip with or without cleft palate [5]. This could be also divided into cleft lip (CL), cleft lip and palate (CLP) and cleft palate (CPO). Cleft palate may influence soft tissues formation and can be also limited to the uvula [4]. Isolated cleft lip affects ca. 15% of all cleft cases, whereas isolated cleft palate affects 40% of the patients [6].

The aim of this narrative review is to focus on the medical assistance that patients will need to be provided throughout their lifetime. It is also important to state, how important the cooperation between many specialists is and how difficult and multifactorial the treatment of these individuals. Patients are affected by a large number of impaired functions, such as problems with breathing, swallowing, chewing, sucking, speaking and hearing [7]. Among the necessary procedures, there are surgeries, orthodontic treatment, laryngologist and speech therapy [8,9]. The oral cavity is a main target for treatment, supporting facial tissues and influencing a patients’ life. The support of genetics and psychology is crucial for patients with clefts nowadays [10]. Many procedures performed by many specialists are needed in cleft patients; therefore, a leader that confers with other specialists would be a perfect solution for cleft patients. The patient requires constant control and referring to other specialist; therefore, the leader should see the patient most often and refer to others when necessary.

The most important questions that should be stated in this research are what the steps for the treatment of patients with clefts are and whether or not there should be a person that coordinates that treatment.

## 2. Inclusion and Exclusion Criteria

Three databases (PubMed, Google Scholar and Cochrane) were searched to obtain the most recent information on the topic of cleft lip and palate treatment. The search criteria included the words cleft lip and/or palate with a combination of the words: multidisciplinary treatment, multidisciplinary approach, surgical treatment, dental treatment, orthodontic treatment and medical care. The articles that were taken into account were peer-reviewed original and review papers published in English. The exclusion criteria assumes that the paper should not be older than 10 years (if possible), and the scientific background should be clear. Figure 1 presents the basic inclusion criteria to conduct that study. The case reports were incorporated into the study only if the novel technique of treatment was introduced in that paper. The aspect of the papers should be practical and have clinical significance. The authors tried to summarize all types of treatment that are performed in patients that require multidisciplinary approach.

Two of the authors (A.P.-S. and M.M.) are workers of a program that establishes burden of care in patients with cleft lip and palate for many years. Both of them are specialists in orthodontics. The aim of this program is to bind doctors of many specialties into the treatment of children with congenital facial deformities, among them clefts. The third author works in diagnostic laboratory and has experience with genetics and in vitro procedures.

Two authors (A.P.-S. and I.D.-I.) searched for the articles. Only the articles that both authors accepted were taken into account when writing this review. The third author (M.M.) accepted the final quantity and quality of the chosen articles and was responsible for the final approval of this paper.

## 3. Newest Treatments of Cleft Lip and/or Palate

### 3.1. Early, Pre-Surgical Treatment

The presurgical procedure should involve presurgical nasoalveolar molding (PNAM also known as Figueroa’s NAM technique). It was firstly introduced by Grayson in 2004 [11]. This reduces cleft stigma, especially around the nasal and lip area. The observation shows reduction of columnella width and improvement of columnella height. It also shows reduction of bi-ala width and increases bi-ala height [12,13]. The pre-surgical preparation may also require a NAM-plate, which is an acrylic orthodontic device to stimulate the maxillary growth and change the growth pattern of the patient with cleft. The novel plates are prepared as a 3D print. NAM plate is helpful with rotation of the premaxilla and reducing the amount of cleft fissure [14]. In addition to those results, the change in the maxillary width in the canine and molar region is also observed [15]. It should be applied as fast as possible due to the possible problems with neonate adaptation. After the preparation, the plate should be worn daily with only a short time for cleaning the plate twice a day [13]. Presurgical preparation also changes the rotation of the incisal bone and therefore the alveolar bone shape becomes less triangular, more natural and the cleft gap spontaneously reduces [12]. The long-term follow-up shows though, that only the shape of the dental arch remains stable in time, other malformations still coexist [16]. The possible ways of mobilizing the soft tissues are lip massage and lip taping in order to make skin more elastic and give it more possibilities to close over the alveolar bone [17,18].

### 3.2. Lip and/orPalate Closure

The preliminary surgical procedures for patients with clefts are cheilo- and palatoplasty, as well as alveolar bone grafting. In most countries, lip and palatal procedures are separated, with different timings 3–6 months for lip and 6–18 months for palatal closure [19,20]. A C-flap technique was recently presented by Jung et al. [18]. This procedure creates a longer lip during the surgery and therefore, the malformation of philtrum and Cupid’s bow peak is lower than with the traditional method. The choice of the method of surgical reparation depends on the surgeon’s experience and the type of the cleft. Most surgeons incorporated Fischer’s technique for the lip repair with modified Millard rotation-advancement flap [21,22,23]. Among many surgical procedures to close the palate, nonradical intravelar veloplasty is one of the methods that might be used to restore it [24].

After the surgery of the lip is performed, the postoperative care should include scar management, massage and observation of the wound potential dehiscence and silicone gel as well as steroid administration [19]. The novel methods of scar management require a laser to reduce the scar tissue by its softening and flattening [25].

### 3.3. Bone Grafting

Most of the patients with cleft of the alveolar bone need bone grafting to restore the shape of the bone for the future teeth movement and prosthetic restoration. The timing of the bone graft is under debate—several techniques of that are used. The most common donor site is iliac crest (most preferable, due to up to 50 mL possible cancellous bone gain) or calvarial bone. The autogenic bone is the best material that could be used in any kind of graft. The bone should be grafted gently, as crushing of the bone increases its resorption and in consequence reduces its volume and quantity [19,26,27,28]. The bone grafting is usually performed at the mixed dentition stage and besides the restoration of collapsed nasal bone, it becomes a base for the ala nasi. The reconstruction of longitude of the alveolar bone makes it possible to restore the teeth loss [7]. Now, the bone grafting should be based on the 3D planning, which enables it to produce actual and fit-to-size bone graft [29]. Additionally, 3D scans might be used for printing scaffolds for bone grafting formed from the bio-glass [8].

## 4. Genetics Changes Underlying the Cleft Lip and/or Palate

The genetic factors in the hereditary of clefts play a crucial role. Among the most common genes in non-syndromic clefts are named: CDH1 (locus 16q22.1), COL2A1 (12q13.11), CRISPLD2 (16q24.1), FOXE1 (9q22.33), GRHL3 (1p36.11), IRF6 (1q32.2), JAG2 (14q32.33), MSX1 (4p16.2), PAX7 (1p36.13), ROCK1 (18q11.1), SUMO1 (2q33.1), TBX22 (Xq21.1), TCOF1 (5q32-q33.1) and TGFA (2p13.3) [4,24]. The genetic sequence could be important for the patient in terms of future reproductive plans. With that knowledge one could plan pregnancy more consciously and the prenatal diagnosis might be crucial in those kids. Knowing that PAX9 and MSX1 genes may predispose to hypodontia, the prenatal diagnostic is also possible in that area [30,31,32]. Further studies regarding genetics of clefts and accompanying dismorphic features could give the doctors wider knowledge on how to treat patients and how to help them in the future. Knowing the genetics of the individuals could help doctors to predict the accompanying problems of the patient and find the best option for their care. If genetic engineering comes into account in the nearest future, the human species could probably exclude the genetic mutation leading to clefts for future individuals. The scientific world discusses the aspects of genetic engineering, but the authors want to be careful with those kinds of statements, especially for the near future.

## 5. Medical Teams Facing the Cleft Lip and/or Palate Treatment and Diagnostics

### 5.1. Pediatrician

The pediatric care of the child with cleft starts at the first diagnostics process. A newborn has difficulties sucking through a regular nipple, due to the lip seal that does not exist. The gap in the lip and in the palate does not let the patient eat normally. Additionally, the gap in the roof of the mouth makes it impossible to produce the build-up of pressure in the mouth. Most of the babies require a special nipple that should be adjusted before leaving the neonatal hospital [6]. The pediatrician could also be a leader of the team, coordinate the other specialist’s work timing, as well as be a support in psychosocial problems, especially when communication problems appear [33].

### 5.2. Logopedics and Phoniatrics

The cleft patients have affected muscle phonation. This dysfunction applies to muscle levator velirud palatini. The most common observation is retardation of consonant sounds (p, b, t, d k, g). The velopharyngeal dysfunction (VPD) is corelated with hypernasality, problems with compensatory articulation and air emission. Nasal resonance and difficulties in articulation are also very common [6,34]. Due to palatal shortage, the nasal speech is present twice more often in cleft toddlers than in healthy individuals [35]. Although more than 60% of the patients with clefts present understandable speech, more than 70% need speech therapy [36].

It should also be considered by the speech therapist and consulted with the laryngologist, whether the nasal emission is obligatory or compensatory, or is it caused by misarticulation. The last phenomenon is observed when pharyngeal sound is substituted by and oral sound. The speech therapist should be able to recognize, whether this condition is a result of abnormal velopharyngeal structure [37]. To obtain longer palatal length, Furlow’s palatoplasty is the method of choice [38]. Primary as well as secondary palatoplasty give the same results [39].

### 5.3. Laryngologist

Laryngological problems usually refer to infections of the middle ear and may result in hearing loss. The biomechanism of otitis media in this situation bases on dysfunction of muscle levator veli palatini, which opens the Eustachian tube [6]. The middle ear effusion may be present at birth and therefore the patient may fail the hearing test [40]. Acoustic rhinometry may be a method of choice when considering the nasal patency, as within the years patients require many surgeries. Acoustic rhinometry helps measure the laryngological problems with the nose and the potential need of the surgical reconstruction [41]. A large number of rhinoplasties are performed on juveniles, the rib grafts are the most common donor sites to restore the nasal shape, length and width. The second most common procedure is nasal osteotomy [42].

### 5.4. Dentists

In patients with cleft lip or palate, dental issues may refer to several from below presented the changes in number of teeth, their shape and time of eruption. The incidence of the neonatal teeth is considered to be common [43]. Microdontic lateral maxillary incisors (or missing ones) are also a very common issue. Another very frequent dental problem is ectopic teeth eruption, specifically referring to the canine that erupts at the palatal side. Delayed tooth eruption and enamel hypoplasia [44] are causes of dental lamina injuries [6,45]. Another dental anomaly is the presence of supernumerary teeth, which affects almost one fourth of the cleft patients and is probably caused by division of dental lamina during prenatal time [46]. The delayed tooth development refers mainly to the cleft side [47]. Dental anomalies refer mostly to the cleft side [48].

Before the reconstructive operation, the patient should be consulted with the pediatric dentist in terms of diet and oral cavity hygiene. After the surgical procedure, topical fluoride and sealants applications should be considered [45]. The fluoride should be administered at least twice a year [40].

It is also a very common issue that the oral hygiene in cleft individuals is improper [45]. Additionally with demarcated or diffused opacity and enamel hypoplasia this may lead to higher predisposition to dental caries [49]. It had been observed that teeth hypoplasia is strongly related to the palatal defect, as well as number and type of performed surgeries [50].

The patients with clefts are more susceptible to caries; therefore, the early preventive advise for caries prevention should be performed. If necessary, the restorative care and endodontic treatment should be performed [45,51]. Children are at a higher risk of developing early childhood caries [40]. Although the recessions are not present, it has been proven that the keratinized gingiva is thinner around the cleft area [52]. The checkups should be common, especially at the stage of active orthodontic treatment with any fixed device, as the hygiene would be more difficult to deal with at this stage [40]. The pediatric dentist could also be another “link” between all members of the multidisciplinary team and a potential leader in treatment planning [45].

#### 5.4.1. Orthodontic Procedures

The most common orthodontic problems refer to malocclusions, which often needs orthognathic surgery in the future. The most frequent ones are crossbites and partial open bites, especially at the surgery side. Among the angle classification, the most frequent are Class I malocclusions, but in difference when compared to the population of healthy individuals, more frequent are Class III malocclusions with dominance of pseudomesioclussion connected to maxillary hypoplasia, whereas, in healthy individuals, Class II malocclusions are three to eight times more frequent. There are also theses that hypoplasia is related to the whole midface complex [53,54,55]. It is also popular to use GOSLON-Yardick scale for defining dentoalveolar malocclusions among patients with clefts. It is a five-category scale that helps divide malocclusions and plan patient’s treatment needs. The GOSLON index is presented in Table 1. GOSLON 3 and higher are observed in ca. 60–70% of patients [56].

The treatment of patients with clefts is a long procedure—most of the patients start their treatment during early childhood, according to Roguzińska at al. [57], at the mean age of 4.83 years and lasts for more than 9 years for unilateral clefts and 4.53 years and over 10 years, respectively, for bilateral clefts. Orthodontic procedures require removable and fixed appliances [57]. Removable appliances are used to gain space for the teeth or keep the space if there was premature loss; therefore, transversal screws are used in those kind of appliances [45].

The novel techniques in orthodontic diagnostics are based on dental scans, which is especially helpful in patients with clefts as the 3D image is more accurate and precise and therefore the individual appliances produced for the patients are expected to fit patients’ dental arch better [58,59].

#### 5.4.2. Prosthodontics

Prosthodontist should be a part of the cleft team. The result of smile attractiveness and naturality depends mainly on the quality of prosthodontic approach. In the past, the majority of dental restorations concentrated around veneers and dental crowns. The issue of prosthodontist is to improve the visual aspect, especially of the damaged and malformed teeth. Dental bridges and removable prostheses were replacing the missing teeth [60,61]. Right now, the most desirable reconstruction of the missing teeth is dental implant. The high survival rate is observed among the dental implants placed in the alveolar bone grafts, although long-term esthetic follow-ups would be desired [62,63]. The pink esthetics due to thinner gingival margin and lower bone levels is worse in individuals with clefts, although the peri-implant parameters remain similar to the control group. The best results were obtained when the bone grafts were performed 3 months before the implant placement [64].

#### 5.4.3. Plastic and Orthognathic Surgery

Plastic and orthognathic surgery pertains to adulthood of the individual. It should include lip revision to correct labial shape and outcome, secondary palatoplasty (correction of velopharyngeal dysfunction), correction of maxillary hypoplasia, rhinoplasty and velopharyngeal incompetence surgery [19]. Due to the characteristic nasal speech of the patients with cleft, their improvement in velopharyngeal function is required to correct them into the non-nasal speech [65,66].

Maxillary hypoplasia is one of the most visible stigmata of the cleft. Oral distractors or orthognathic surgery should cause less visibility in the cleft malformations [19]. Osteotomy (most frequently Le Fort I) is performed to improve the patient’s life quality and facial appearance. It should be combined with bone grafting to avoid velopharyngeal function (VPF) disturbances, but still, the patients should be informed before the Le Fort I osteotomy that they may require more surgeries to improve their speech [67]. Most of the patients would be willing to have a rhinoplasty simultaneously when the Le Fort procedure is performed [9]. In patients with clefts, the usual revision rhinoplasty requires cartilage grafts [68].

## 6. Other Treatment Needs

The facial deformation, congenital pain and need for of repair procedures may influence a patient’s life [69,70]. The will to look better and nicer is the psychosocial problem of individuals with cleft. The need for acceptance pushes these individuals to put posts on the internet with positively stigmatized words such as beauty [71]. Therefore, the need for psychologist might be crucial. It has been shown, though, that improvement of smile and occlusion with orthodontic treatment act the same when compared to healthy individuals; there is no difference in life quality among those two groups [72].

Patients with clefts present masticatory muscle function overload, which may require orthodontic and prosthetic treatment as well as physiotherapy [73,74,75]. The ideal treatment planning to restore the proper bite, intercuspidation and function would require using facebows and articulators in order to not overload the temporomandibular joint and plan the equal pressure on all points on the teeth [66].

## 7. Discussion

Patients with clefts require multidisciplinary treatment throughout all of their lives. Knowing the overall problems of orofacial deformities in patients with clefts may result in the expansion of the need for multidisciplinary treatment. The most common malocclusions present in cleft patients are crossbites and open bites, especially on the cleft side. The problem of malocclusions lies in the underdevelopment of hypoplastic maxilla [53]. Although the patient requires several reconstruction surgeries (lip reconstruction, palatal closure and bone grafts) and plastic surgeries to improve the overall result, the result of the treatment lies in orthodontic-orthognatic preparation in most of the cases. Lacking the palatal suture and presence of the scar on the palate rotates the palatal bone mesially, especially in the frontal region. The patient requires maxillary segmentalization to improve the stability of maxillary complex after the surgical, orthognathic procedures in adulthood [53,76]. It is also important to establish vertical and sagittal relation of the jaws, as it gives a higher knowledge on the 3-dimensional problem of malocclusion [77].

In addition to reconstruction and surgical procedures, patients with clefts require many specialists on their way to the “normality”. Speech therapists, laryngologists and phoniatrics take care of the development of speech and loss of nasality typical for cleft patients [34,35,36,37,38,39,40,41,42]. Due to several dental anomalies that are present in the patients, those patients require additional multidisciplinary dental care, including a child dentist, orthodontist and finally, in most of the cases, prosthodontist to achieve symmetrical and the most natural effect that is possible [78,79].

The team that takes care of the patient with cleft should cooperate with each other very strongly. In the authors opinion, creating the programs of orthodontic care burden, as one in our country, is a great idea to create a web of specialists is extremely beneficial. Creating the schemes of the treatment would keep the patient and the parents well-informed and makes it easier to cooperate between the specialists of all the medical specialties. It would also be easier to incorporate the novel treatment methods to those patients in a cohesive and cooperative team. It is also easier to refer the patient to the specialist of other disciplines in that case. In the authors opinion, it is mandatory to have a leader in this kind of team, although it would be disputable who it should be. The main goal of the leader would be creating a kind of web and bind other specialists when the care is applied to the patient. The cleft patients require specialists that cooperate, but also have high experience in this topic. The “evidence-based” medicine shows that the greater the experience of the doctor, results in the best possible outcomes. This also applies to that topic.

## 8. Conclusions

To sum up, clefts are very common birth defects that require multidisciplinary approach. Clefts require many surgeries—reconstructive and, later, plastic ones—to restore the continuity of tissues and impaired functions. The first surgeries take place in the neonatal period and the whole treatment is long-term, requires many specialists and lasts until adulthood. Due to the most frequent and systematic checkups (at least twice a year), the leader of the team, which refers patient to the other specialist, in the authors mind, could be a pediatrician, dentist or orthodontist.

## 9. Limitations

The first limitation of this study is incorporating only three databases into the search criteria. The authors decided to do that because of the nature of this research, knowing that neither a systematic review nor meta-analysis could be carried out on this topic. The authors are aware that this topic could be widened, and each section of this review could be treated as a separate paper.

The experience of the authors of this study, based on the dental and orthodontic knowledge, mainly due to the fact that it is the main area of expertise of the authors. Working with the cleft patients, we made some observations regarding the schemes of treatment in general and tried to present it.

## Figures and Tables

**Figure 1 jpm-12-02089-f001:**
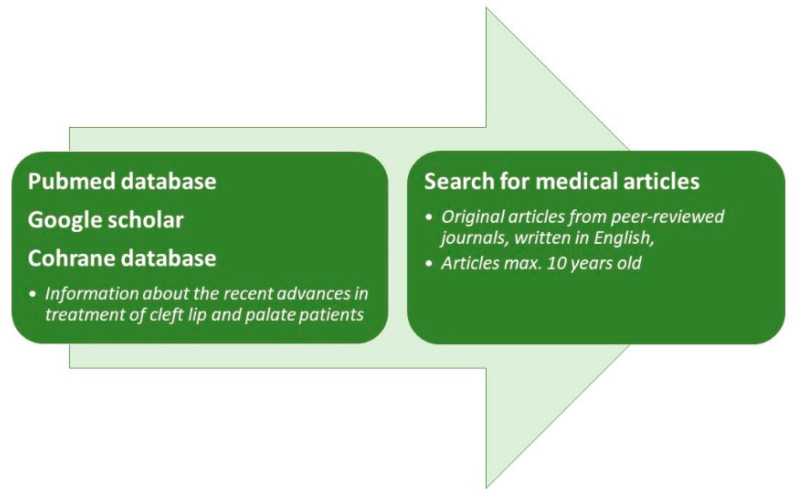
The inclusion criteria of papers incorporated to the study.

**Table 1 jpm-12-02089-t001:** GOSLON Index—the division, description and treatment needs [53,56,57].

GOSLON Scale	Meaning	Treatment Needs
GOSLON 1	Isolated dental anomalies	Orthodontic treatment could be performed due to esthetic reasons
GOSLON 2	Lateral crossbite, palatotrusion of upper incisors	Removable and fixed appliance treatment
GOSLON 3	Lateral crossbite, tête-à-tête. occlusion of the incisors; possible complete crossbite of one half of the arch	Removable and fixed appliance treatment with additional transpalatal bars and hyrax appliance
GOSLON 4	Severe malocclusion with crossbite on the bone basis	Orthognathic surgery; MARPE
GOSLON 5	Severe malocclusion with crossbite on the bone basis with additional open bite	Orthognathic surgery

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
