# Peer review of "Current Concepts and Challenges in the Treatment of Cleft Lip and Palate Patients—A Comprehensive Review"

_jpm, 2022, doi:10.3390/jpm12122089_

Round 1
Reviewer 1 Report
The authors reviewed overall treatment of cleft lip/palate in this paper.
The authors tried to provide the latest knowledge and issue about the management of cleft lip/palate, but the contents of manuscript was general information and less interesting for readers. Furthermore, the aim of this paper is too broad to contain the details of current status of cleft lip/palate management. For example, there are many surgical modification in CL/P surgery and C-flap technique is not a novel technique for cleft lip repair as author mentioned. Most surgeons adopted Fisher's anatomical subunit technique for cleft lip repair and radial IVV for cleft palate repair as a novel technique. In alveolar bone grafting, there are several issues to debate, such as surgical timing, bone substitutes.
Although they tried to provide a lot of information to the readers, I think it would be helpful to the readers to provide in detail on a specific topic.
Author Response
Dear Reviewers,
Thank you for the effort in reviewing our paper. We submitted it after the corrections. We also asked a native speaker to check the whole manuscript and correct language style. Here are detailed response to the two reviews:
Reviewer 1
The authors reviewed overall treatment of cleft lip/palate in this paper.
Dear Reviewer, thank you. That was acctually the goal of our study
The authors tried to provide the latest knowledge and issue about the management of cleft lip/palate, but the contents of manuscript was general information and less interesting for readers. Furthermore, the aim of this paper is too broad to contain the details of current status of cleft lip/palate management. For example, there are many surgical modification in CL/P surgery and C-flap technique is not a novel technique for cleft lip repair as author mentioned. Most surgeons adopted Fisher's anatomical subunit technique for cleft lip repair and radial IVV for cleft palate repair as a novel technique. In alveolar bone grafting, there are several issues to debate, such as surgical timing, bone substitutes.
Thank you for the comment. The authors of this paper are working in the childcare burden programme for several years each. We tried to incorporate more detained information of all the topics, but we know that each paaragraph could be a separate paper. We tried to correct the surgical methods of treatment to the comments provided by the Reviewer. Our goal was not going deep into surgical procedures though. Of course, the bone grafting is disputable, the same as timing of the lip and palate closure. This is a perfect idea for another type of manuscript, preferabely a metaanalysis provided by the experienced surgeon.
Although they tried to provide a lot of information to the readers, I think it would be helpful to the readers to provide in detail on a specific topic.
Thank you for the comment. We also think that it would be interesting to have more papers, each focusing on the separate chapter. Our goal was to show how complicated it is to take care of the cleft patients – it needs a lot of cooperation between the people of many specialties, therefore we did not focus on the one and speciffic type and method of treatment.
Reviewer 2 Report
The manuscript entitled "Current concepts and challenges in the treatment of cleft lip 2 and palate patients – a comprehensive review." has an interesting and useful purpose and aim.
The introduction is focused on this major problem and points out the most significant general aspects correlated with the domain.
Inclusion criteria: why is only PubMed database included? There is also Cochrane and google scholar mentioned in the figure, but not in the text. What about Medline and WoS? What were the key word for the search?
4. Genetics changes underlying the cleft lip and / or palate- this should /could be implications for future studies and treatments. Just mentioned like this, alone, doesn't have special value in the manuscript
3. Newest treatments of cleft lip and / or palate- should be after all previously mentioned as standard treatments
Discussion section lacks discussion of the challenges that the multidisciplinary team faces in the treatment of this group of patients.
The conclusion did not give a precise answer to the aim of the study: "The most important question that should be stated in this research is what the steps of the treatment of patients with clefts and whether are or not there should be a person, that coordinates that treatment." please revise conclusion according to this question.
References 54, 55, 59 are older then 10 years. (as search exclusion parameter) Also, references need revision according to the citing propositions of the journal.
Author Response
Dear Reviewers,
Thank you for the effort in reviewing our paper. We submitted it after the corrections. We also asked a native speaker to check the whole manuscript and correct language style. Here are detailed response to the two reviews:
Reviewer 2
The manuscript entitled "Current concepts and challenges in the treatment of cleft lip 2 and palate patients – a comprehensive review." has an interesting and useful purpose and aim.
Dear Reviewer, Thank you for a nice support
The introduction is focused on this major problem and points out the most significant general aspects correlated with the domain.
That was the goal of our research.
Inclusion criteria: why is only PubMed database included? There is also Cochrane and google scholar mentioned in the figure, but not in the text. What about Medline and WoS? What were the key word for the search?
Thank you for the comment and sorry for the inconvenience, that was caused by searching in the first database and than extending it to two more. The authors added the missing information to the main text. We also added the information on the search criteria in the text. Due to it is not a systematic review or metaanalysis (the type of the study does not provide this type of manuscript) but a comprehensive review, the search criteria were limited by authors to those databases. Due to Authors experience from the orthodontic / dental point of view, the care burden was presented from our long-term experience in that area and could be incomplete. Therefore we decided to add that issues as a separate chapter – limitations.
- Genetics changes underlying the cleft lip and / or palate- this should /could be implications for future studies and treatments. Just mentioned like this, alone, doesn't have special value in the manuscript
Thank you for the comment. The authors widened this chapter to issues how could probably the genetic information be used. We also added the information on the probability of incorporation of genetic ingeneering, although we are very cautious and conservative in that topic and would not like to widened it, as to authors mind, this is probably not the nearest future of the medicine.
- Newest treatments of cleft lip and / or palate- should be after all previously mentioned as standard treatments
Discussion section lacks discussion of the challenges that the multidisciplinary team faces in the treatment of this group of patients.
Thank you for the comment, we tried to correct that and add the missing information and widen that topic
The conclusion did not give a precise answer to the aim of the study: "The most important question that should be stated in this research is what the steps of the treatment of patients with clefts and whether are or not there should be a person, that coordinates that treatment." please revise conclusion according to this question.
Thank you for the comment, we tried to correct that and add the missing information and widen that topic
References 54, 55, 59 are older then 10 years. (as search exclusion parameter) Also, references need revision according to the citing propositions of the journal.
Thank you for the suggestion, we changed the references 54 and 55 to the novel ones. Reference 59 comes from the past 10 years. We also added one more reference to that topic
Round 2
Reviewer 1 Report
Thank you for kind response to my review
I think it will be helpful article to the beginner cleft lip/palate surgeon by concisely summarizing overall treatment of cleft lip/palate management.
Congratulation!
Reviewer 2 Report
The manuscript is corrected and good for publication.